# Genomic Characterization of *Escherichia coli* Isolates Belonging to a New Hybrid aEPEC/ExPEC Pathotype O153:H10-A-ST10 *eae*-beta1 Occurred in Meat, Poultry, Wildlife and Human Diarrheagenic Samples

**DOI:** 10.3390/antibiotics9040192

**Published:** 2020-04-17

**Authors:** Dafne Díaz-Jiménez, Isidro García-Meniño, Alexandra Herrera, Vanesa García, Ana María López-Beceiro, María Pilar Alonso, Jorge Blanco, Azucena Mora

**Affiliations:** 1Laboratorio de Referencia de Escherichia coli (LREC), Departamento de Microbioloxía e Parasitoloxía, Facultade de Veterinaria, Universidade de Santiago de Compostela (USC), 27002 Lugo, Spain; dafne.diaz@usc.es (D.D.-J.); isidro.garcia@usc.es (I.G.-M.); alex.herrera.est@gmail.com (A.H.); vanesag.menendez@usc.es (V.G.); jorge.blanco@usc.es (J.B.); 2Instituto de Investigación Sanitaria de Santiago de Compostela (IDIS), 15706 Santiago, Spain; 3Department of Veterinary and Animal Sciences, Faculty of Health and Medical Sciences, University of Copenhagen, 1870 Frederiksberg, Denmark; 4Departamento de Anatomía, Produción Animal e Ciencias Clínicas Veterinarias, Facultade de Veterinaria, Universidade de Santiago de Compostela (USC), 27002 Lugo, Spain; anam.lopez.beceiro@usc.es; 5Unidade de Microbioloxía, Hospital Universitario Lucus Augusti (HULA), 27003 Lugo, Spain; Pilar.Alonso.Garcia@sergas.es

**Keywords:** *Escherichia coli*, ESBL, hybrid pathotype, ExPEC, EPEC, MDR, ST10, O153, EnteroBase

## Abstract

Different surveillance studies (2005–2015) in northwest Spain revealed the presence of *eae*-positive isolates of *Escherichia coli* O153:H10 in meat for human consumption, poultry farm, wildlife and human diarrheagenic samples. The aim of this study was to explore the genetic and genomic relatedness between human and animal/meat isolates, as well as the mechanism of its persistence. We also wanted to know whether it was a geographically restricted lineage, or whether it was also reported elsewhere. Conventional typing showed that 32 isolates were O153:H10-A-ST10 *fim*H54, *fimAv_MT78_*, *traT* and *eae*-beta1. Amongst these, 21 were CTX-M-32 or SHV-12 producers. The PFGE *XbaI*-macrorestriction comparison showed high similarity (>85%). The plasmidome analysis revealed a stable combination of IncF (F2:A-:B-), IncI1 (STunknown) and IncX1 plasmid types, together with non-conjugative Col-like plasmids. The core genome investigation based on the cgMLST scheme from EnteroBase proved close relatedness between isolates of human and animal origin. Our results demonstrate that a hybrid MDR aEPEC/ExPEC of the clonal group O153:H10-A-ST10 (CH11-54) is circulating in our region within different hosts, including wildlife. It seems implicated in human diarrhea via meat transmission, and in the spreading of ESBL genes (mainly of CTX-M-32 type). We found genomic evidence of a related hybrid aEPEC/ExPEC in at least one other country.

## 1. Introduction

*Escherichia coli* is a normal inhabitant of the human and animal intestinal tract. However, *E. coli* can also act as a pathogen in a broad range of conditions, from enteric diseases to extraintestinal infections, such as urinary tract infection (UTI) and sepsis, among others. Strains that cause enteric disease or diarrhea are called diarrheagenic *E. coli* (DEC), which includes six major categories characteristically defined by certain pathotype-specific virulence markers [1,2]. Thus, the enteropathogenic *E. coli* (EPEC) category is typically carrier of the *eae* gene, as part of the pathogenicity island locus of enterocyte effacement (LEE), codifying a protein called intimin. The intimin is responsible for the intimate adherence of the bacteria to the enterocyte membranes and, eventually, for the attaching and effacing (AE) lesion of the brush-border microvilli [3]. The variable C-terminal-encoding sequence of *eae* defines more than 30 distinct intimin types and subtypes associated with tissue tropism [4,5]. EPEC are further classified as typical (tEPEC) when they carry an EPEC adherence factor (EAF) plasmid that encodes adherence mediated by the bundle forming pilus (BFP), while atypical EPEC (aEPEC) produce the AE lesion, but do not express BFP [4,6]. Currently, aEPEC isolates are emerging enteropathogens detected worldwide and isolated from different niches (animal species, environment, and food samples), while the main reservoir of tEPEC isolates are humans [7,8]. On the other hand, *E. coli* that cause extraintestinal infections are referred to as extraintestinal pathogenic *E. coli* (ExPEC), which includes a heterogeneous group of pathotypes defined by isolation from infections outside the intestinal tract: uropathogenic *E. coli* (UPEC), avian pathogenic *E. coli* (APEC), and neonatal meningitis *E. coli* (NMEC). A variety of specific virulence genes have been described in ExPEC (adhesins, protectins, siderophores, toxins, hemolysins, polysaccharide antigens, invasins, colicins, etc.) [1,2]. However, even though certain virulence traits and phylogenetic groups have been proposed to characterize ExPEC [9,10], no set of genes can be used to unequivocally distinguish them from commensal *E. coli*. In fact, ExPEC strains can commensally colonize the human intestine, which in turn, can serve as reservoir [11]. Antimicrobial resistance is a serious global concern which involves the health care system, food production and environmental integrity [12]. In fact, it is assumed that antimicrobial drugs used in the livestock sector play an important role in the spread of extended-spectrum beta-lactamases (ESBL)-producing *E. coli* throughout the food chain to humans [13,14]. The genomic plasticity of *E. coli* is the consequence of the important role played by mobile genetic elements (MGEs) such as plasmids, bacteriophages, pathogenicity islands, transposons and insertion sequence elements in the evolution of the bacteria [15]. As a result, hybrid *E. coli* pathotypes unpredictably emerge, given the mobility of most of the genes encoding virulence and antimicrobial resistance (AMR) [15,16]. Since 2011, when a novel Shiga-toxin-producing *E. coli* (STEC) belonging to serotype O104:H4, with virulence features (VF) common to the enteroaggregative *E. coli* (EAggEC ) and CTX-M-15 producer was identified as the one involved in the large German outbreak [17], the concept of pathotype has been questioned. Currently, classical and new approaches, such as whole genome sequencing (WGS), are being used to enhance the understanding of the evolution of this highly adaptable species [16,18].

Different in-house surveillance studies in northwest Spain (2005-2015) revealed the presence of *eae*-positive isolates of *E. coli* O153:H10, many of them ESBL-producers, in meat for human consumption, wildlife, and avian farm environments. We also found them involved in human diarrhea. The aim of this study was to explore the genetic and genomic characterization relatedness between human and animal/meat isolates, as well as to gain knowledge in the mechanism that might be playing a role in its persistence. We also wanted to know whether it was a geographically restricted *E. coli* lineage, or whether it was also reported somewhere else.

## 2. Results

Thirty-two *eae*-positive *E. coli* (21 ESBL and 11 non-ESBL) belonging to the serotype O153:H10 constituted the collection of the study: 14 from human stools, 8 from beef meat, 7 from chicken meat, and 1 each from pork meat, wildlife (fox feces) and a poultry farm environment sample, as shown in Appendix A. They were detected within different surveys from 2005 to 2015 based on the following findings. First, we noticed the recovery of *eae*-positive isolates with the same intimin type. Second, many of them were ESBL-producers (coming from surveillance studies on ESBL-producing *E. coli* in animal sources). Third, they belonged to the O153 serogroup. Fourth, we found them within the human collection in the retrospective analysis. Fifth, the isolates also carried ExPEC virulence genes. 

### 2.1. Conventional Typing

Table 1 summarizes the main traits obtained by conventional typing for the 32 isolates. All were positive for the intimin *eae-*beta1, but negative for *bfpA* gene, conforming the aEPEC pathotype. Other virulence genes defining verotoxigenic (VTEC), enteroinvasive (EIEC), enteroaggregative (EAggEC) or enterotoxigenic (ETEC) pathotypes were not detected. Interestingly, the investigation of virulence traits associated with ExPEC showed that the *fimAv_MT78_* gene, which is a virulence *locus* that codify a *fimA* variant MT78 of type 1 fimbriae [19], was present in all isolates. In addition, the *traT* gene that codifies an outer membrane protein implicated in serum survival [20] was also determined in 17 of the isolates, as shown in Table 1. Finally, all the isolates were assigned to the clonal group O153:H10-A-ST10 (CH11-54) by means of the serotyping, phylogroup, MLST and clonotyping.

The highest rates of AMR were to: ampicillin (75%; 24/32), cefuroxime (68.7%; 22/32), cefotaxime (65.6%, 21/32), ceftazidime (65.6%, 21/32), cefepime (59.4%, 19/32) and gentamicin (59.4%, 19/32). The ESBL-typing of the 21 positive isolates determined that 19 were CTX-M-32 and two SHV-12 producers, as shown in Table 1. 

The comparison of the *XbaI*-macrorrestriction profiles obtained by pulsed field gel electrophoresis (PFGE) of the ESBL-producing aEPEC isolates revealed high similarity. Thus, all but one clustered with an identity >85% in the dendrogram shown in Figure 1. It is of note that three human clinical isolates, recovered in different years, clustered each with a fox (95.2% of similarity) and with two beef meat isolates (100% and 97.6% of similarity, respectively).

### 2.2. Whole Genome Sequencing (WGS)

Based on the high similarity shown by PFGE and to further investigate the virulence profile, resistome, plasmid content and relatedness, 17 representative aEPEC/ExPEC isolates were WG sequenced. The de novo assembled contigs were typed in silico using the EnteroBase tools, as shown in Appendix A, as well as the Center for Genomic Epidemiology (CGE) databases, also shown in Table 2.

SerotypeFinder and EnteroBase predictions corroborated O and H antigens, with the exception of LREC-120 and LREC-121, for which O153 was solved by serotyping. MLST (CGE and EnteroBase), CHTyper and ClermonTyping also confirmed conventional data for ST (10), CH (11-54) and phylogroup (A), shown in Table 2 and Appendix A. Additionally, the wgST, cgST, and rST of the genomes were determined using the schemes of EnteroBase, shown in Appendix A. Whole genome multilocus sequence typing (WgMLST) and core genome multilocus sequence typing (cgMLST) are powerful schemes with extreme and high resolution, respectively. cgMLST is defined as MLST based on the core genome, whereas wgMLST is based on a non redundant set of genes across a species, similar to a ‘pan-genome’. In EnteroBase, the number of core loci included in the cgMLST scheme for *E. coli* is 2,512 and 25,002 in the wgMLST. Different cgSTs and wgSTs were assigned to each of the 17 genomes analyzed, while rST (medium resolution; 53 loci) was the same (2021) for all genomes, excluding LREC-127 (58738), as shown in Appendix A.

VirulenceFinder corroborated the hybrid pathotype nature of the isolates, predicting in all genomes the *eae* gene (intimin) together with other components encoded in the LEE pathogenity island, as well as the increased serum survival gene *iss* recognized for its role in ExPEC virulence [21]. The *astA* gene, which encodes the heat-stable enterotoxin 1, was also present in all 17 isolates (Table 2).

ResFinder identified the genes associated to resistances observed in vitro (acquired resistances for beta-lactams, aminoglycosides, and point mutations for quinolones), only, the *bla_CTX-M-32_* was not predicted in silico for LREC-112 and LREC-119, but by conventional sequencing. Furthermore, ResFinder determined other acquired resistances which had not been tested in vitro, such as to phenicols and macrolides in all genomes, and to tetracyclines in 16 out of the 17 genomes, as shown in Table 2. 

Based on the replicon identification, PlasmidFinder revealed a homogenous profile of four or five plasmid types. Thus, the concomitant presence of IncF (F2:A-:B-), IncI1 (STunknown) and IncX1, together with non-conjugative Col156-like plasmids, was detected in 15 of 17 genomes. Four of those 15 genomes were also carriers of Col (MG828)-like plasmids, as shown in Table 2.

In the asymmetric distance matrix on the cgMLST scheme from EnteroBase, based on the presence or absence of 2,513 genes, the 17 genomes showed <20 differences (range 5–19) in relation to the human diarrheagenic isolate LREC-113, as displayed in Table 3 and Figure 2. We also looked into the static Hierarchical Clustering (HierCC) designations in EnteroBase. The 17 genomes were assigned into the same HierCC HC50 (37600), which means all strains in this cluster have links no more than 50 alleles apart. Furthermore, using HC20, three human genomes (LREC-113, LREC-116, LREC-124) and two beef meat (LREC-119, LREC-125) clustered together (37606) with links no more than 20 alleles apart, as shown in Appendix A. A dendrogram was also built in EnteroBase based on the SNPs of the core genomic regions present in 90% of the compared genomes, and using LREC-113 as reference, downloaded, and modified with FigTree v1.4.3 as shown in Figure 3. Within 1,068 variant sites, the number of SNPs was <62 for 13 of the 17 genomes, as displayed in Appendix A.

## 3. Discussion

The recovery from different sources of *eae*-positive *E. coli* isolates of serotype O153:H10, and its association with ESBL enzymes triggered this investigation. In independent studies on ESBLs in our region, we found that O153 aEPEC represented 5.5% of the ESBL-producing *E. coli* recovered from chicken meat (2009-2010), 7.7% of pork meat (2011-2012), 20% of beef meat (2011-2012), 1% of poultry farm environments (2010-2012) and 1% of wildlife feces (2014-2015) [22]. In addition, we detected 23 (0.24%) O153 aEPEC as the only pathogen within 9,523 stools of epidemiologically unrelated patients (2006–2012) in the routine testing of human diarrheagenic samples. From those 23, 14 (0.15%) were O153:H10 *eae-*beta1 *fim_AvMT78_*, and five of them CTX-M-32 producers, as shown in Appendix A and Figure 1. By conventional typing, all animal and human isolates were assigned to the clonal group O153:H10-A-ST10 (CH11-54), conforming a hybrid aEPEC/ExPEC pathotype not previously described. The symptomatology reported in humans was mainly mild diarrhea, but there were also some cases of acute and hemorrhagic gastroenteritis, as shown in Appendix A. Epidemiological studies have indicated that aEPEC are emerging enteropathogens, implicated in human diarrhea, with higher prevalence than tEPEC in both developed and developing countries [23]. aEPEC are present in both healthy and diseased animals and humans [8,24,25], are phylogenetically heterogeneous, and carry virulence factors of other diarrheagenic *E. coli* more often than tEPEC strains [6,23,26]. However, the main feature of the EPEC diarrheagenic group is the ability to induce A/E lesions on intestinal epithelium encoded in the chromosomal pathogenicity island LEE. Within more than 30 intimin types and subtypes based on the polymorphism of *eae*, the subtype determined here (beta-1) is first or second in prevalence within different studies on isolates from humans with diarrhea in Spain [24], Australia [27], Brazil [28,29], Peru [30] or China [31]. 

It is of note that in subsequent and current studies, we have detected this clonal group on meat samples from supermarkets in our city. In fact, we recovered aEPEC/ExPEC from 15 out of 100 poultry meat samples (2016-2017), and of those, five were carriers of isolates belonging to the clonal group O153:H10-A-ST10, including one CTX-M-32 producer (unpublished data). Recently, Zhang et al. [32] reported a 2.75% prevalence of aEPEC in retail foods at markets in the People’s Republic of China, with the beta-1 intimin and the ST10, as the second intimin and ST most prevalent within their isolates. According to the authors, the presence of virulent and MDR aEPEC in retail foods poses a potential threat to consumers. While O153 is a serogroup reported within ExPEC and DEC isolates, and linked to different lineages [33,34,35,36], there are few references of the serotype O153:H10. Notheworthy, the report by Schremmer et al [37] of an aEPEC O153:H10 isolated from the small intestine of a cockatiel (psittaciform bird) with enteritis.

Since the occurrence of the major outbreak of HUS in Europe in 2011 caused by an EAggEC/STEC O104:H4, other hybrid pathotypes have been recognized and new ones are expected, either by novel assemblies of *E. coli* virulence determinants or through the acquisition of new virulence genes from other bacterial species [16]. In Norway, Lindstedt et al. [38], expressed their concern regarding the detection of *E. coli* from human fecal content with a combination of intestinal and ExPEC virulence genes (IPEC/ExPEC) in a high frequency (64.3%). Several other studies have also identified STEC- and ETEC-associated virulence genes coexisting in *E. coli* isolates from humans, animals or environmental origin [39,40]. However, one of the most outstanding is the EPEC/STEC O80:H2-ST301, which emerged in France over the last few years and diffused within Europe. This emerging hybrid is associated with invasive infections, and combines intestinal VFs (*stx2d*, *eae-*xi and *ehxA* genes) and extraintestinal genes characteristic of the plasmid pS88 [41,42]. In this O80 clone, it is to highlight the location of MDR and pS88 genes in the same plasmid, as well as the presence of two additional plasmids (a carrier of *ehxA* gene and a cryptic one) within the isolates [41,42]. The clonal group described here poses also the threat of being MDR and characteristically associated with ESBL-type CTX-M-32. The CTX-M-32 enzyme is derived from CTX-M-1 by a single amino acid replacement, probably an ancestor among CTX-M-1 and CTX-M-15 [43]. The *bla*_CTX-M-32_ gene was first described in 2004 in an *Escherichia coli* isolate in our Health Area (A Coruña, northwest Spain) [43]. Furthermore, it was described in three human isolates O25b:H4-ST131 *ibeA*-positive of our region, as early as in 2008 [14]. Of the 2,427 *E. coli* bloodstream isolates recovered in the hospital of our city (HULA) in the period 2000–2011, 96 were positive for ESBL production, from which 4.2% were CTX-M-32 and 4.2% SHV-12 [44]. The same prevalence was observed in this hospital in 2015 (unpublished data).

The in silico analysis of 17 representative genomes O153:H10-A-ST10 corroborated the main traits determined by conventional typing. In a recent study, we proved the good correlation and usefulness of SerotypeFinder or EnteroBase predictions [25,45]. Here, only the serotype of two genomes could not be predicted in silico, probably due to the limitation of the assembly based on Illumina short reads [46]. MLST, CHTyper from CGE and EnteroBase also confirmed conventional results. Like in the previous study, we found that VirulenceFinder verified the *E. coli* pathotypes (aEPEC/ExPEC) established by PCR. However, VirulenceFinder identified different traits for the ExPEC pathotype. Thus, this clonal group O153:H10-A-ST10 typically carries the locus encoding a *fimA* variant MT78 of type 1 fimbriae [19], and the *traT* gene for an outer membrane protein implicated in serum survival [20]. Both VFs are not included in the VirulenceFinder scheme and so, they are not predicted. On the contrary, the CGE tool allowed the identification of the increased serum survival gene *iss*, recognized for its role in ExPEC virulence [21], in all genomes. The CGE database predicts 14 variants of the *iss* gene [47], including the one described in *E. coli* IAI1 (CU928160), and harbored by the O153:H10-A-ST10 genomes. Our specific PCR detects the plasmid-borne *iss* allele (designated type 1), which is highly prevalent among avian pathogenic *E. coli* and neonatal meningitis-associated *E. coli* isolates, but not among uropathogenic *E. coli* isolates [21]. The phenotypic AMR determined in vitro correlated with the results predicted by ResFinder databases, with the exception of the *bla*_CTX-M-32_ gene not predicted in two genomes, but solved by PCR and sequencing. Based on this, and previous studies [45,48], we consider both conventional and genomic-based analysis complementary for a better understanding and characterization of emerging isolates. 

An important trait found in this study was the concomitant presence of IncF (F2:A-:B-), IncI1 (STunknown) and IncX1, together with non-conjugative Col156-like plasmids in 15 of the 17 genomes. Although carriage of plasmids places a fitness burden on their host [49], different studies support the hypothesis that interference between conjugative plasmids may reduce fitness cost by decreasing the efficiency of transfer. However, the mechanisms of such inhibitory systems need further investigation [50]. On the other hand, small plasmids were shown to increase their stability in cells containing big plasmids [49]. We hypothesized here that the stable combination of IncF (F2:A-:B-), IncI1 (STunknown) and IncX1 plasmid types, together with non-conjugative Col-like types might be implicated in the successful persistence of this hybrid pathotype, and the spread of antibiotic resistance genes. 

Another objective in this study was to know if this was a geographically restricted genetic lineage, or if it had been reported somewhere else. For this purpose, and based on the HierCC Cluster ID, we searched related genomes uploaded in EnteroBase. As a result, we found a hybrid aEPEC/ExPEC pathotype A-ST10 *eae*-beta1 on its database associated to five humans, one avian, and one unknown isolates, as shown in Appendix A. Of note are the two human O153:H10-A-ST10 (CH11-54) *eae*-beta1 isolates (Code Name: 853984 and 866428) from the United Kingdom, which clustered with the 17 Spanish genomes in the HC100 HierCC group (37600), as shown in Appendix A. Additionally, the in silico analysis of these two genomes showed that they were MDR, and carried similar virulence traits (conforming hybrid aEPEC/ExPEC pathotype) and plasmid combination: IncF (F2:A-:B-), IncX1 and Col156-like, as shown in Appendix A. Importantly, six of the seven hybrid aEPEC/ExPEC genomes found in EnteroBase were carriers of IncF (F2:A-:B-) and Col156-like plasmids, as shown in Appendix A. As above suggested, further investigation on the interplay between these plasmids and other mobile genetic elements affecting their transmission and persistence, as well as their role in the maintenance and acquisition of resistance genes is necessary.

## 4. Materials and Methods

### 4.1. E. coli Collection

From 2005 to 2015, different surveillance studies performed at the Reference Laboratory of *Escherichia coli* (LREC), in Lugo, Spain, aimed to detect ESBL-producing *E. coli* within different sources of our region. These studies included samples from chicken, beef and pork meat, as well as poultry farm environments (avian feces taken from the floor) and wildlife (fox feces). Briefly, the confluent growth of the MacConkey Lactose plates from each sample was screened by PCR for the presence of specific *bla* genes using the TEM, CIT, SHV, CTX-M-1 and CTX-M-9 group-specific primers [51]. Then, up to ten individual colonies from positive plates were reanalyzed. Those confirmed for the *bla* genes were further characterized by PCR for the presence of VF *eae*, *stx1*, *stx2*, *ipaH*, pcDV432, *eltA*, *estA* or *estB* associated with the main intestinal pathotypes (enteropathogenic, verotoxigenic, enteroinvasive, enteroaggregative and enterotoxigenic) of *E*. *coli*. Likewise, specific extraintestinal VF were tested: *fimH*, *fimAv*_MT78_, *papC*, *sfa/focDE*, *afa/draBC*, *cnf1*, *cdtB*, *sat*, *hlyA*, *iucD*, *iroN*, *kpsM II* (establishing *neuC*-K1, K2 and K5 variants), *kpsM III*, *cvaC*, *iss*, *traT*, *ibeA*, *malX*, *usp* and *tsh* (Appendix A). 

On the other hand, human diarrheagenic *E. coli* isolates, mainly from the Hospital Universitario Lucus Augusti (HULA) of our city (Lugo, northwest Spain), were routinely analyzed in our laboratory for intestinal VF, and those positive, complementary analyzed for extraintestinal traits and ESBL genes, as described in the preceding paragraph.

All isolates were serotyped using the method previously described by Guinée et al. [52] employing O1 to O185 and H1 to H56 antisera. As a result, 32 *eae*-positive *E. coli* (21 ESBL and 11 non-ESBL) belonging to the serotype O153:H10 constituted the collection of the study, as shown in Appendix A.

### 4.2. Antimicrobial Susceptibility and ESBL Typing

Antimicrobial susceptibility testing was conducted by disk (Becton Dickinson, Sparks, MD, USA) diffusion assay. The antibiotics tested included ampicillin (AMP), amoxicillin/clavulanic acid (AMC), cefuroxime (CXM), ceftazidime (CAZ), cefotaxime (CTX), cefepime (FEP), cefoxitin (FOX), aztreonam (ATM), imipenem (IMP), gentamicin (GEN), tobramycin (TOB), fosfomycin (FOF), nitrofurantoin (NIT), sulfamethoxazole/trimethoprim (SXT), ciprofloxacin (CIP) and nalidixic acid (NAL). The assays were performed, and all results interpreted, according to the CLSI guidelines [53]. The isolates were investigated by PCR for screening of specific *bla* genes using the TEM, SHV, CTX-M-1 and CTX-M-9 group-specific primers (Appendix A), and further sequencing as described elsewhere [51].

### 4.3. Phylogenetic Assignment and PFGE Comparison

Phylogroup and ST assignment was performed following the Clermont et al. [54] and Achtman MLST [55] schemes, respectively. The clonotyping was based on the internal 469-nucleotide (nt) and 489-nt sequence of the *fumC* and *fimH* genes, respectively, to define the CH type [56]. The molecular similarity within the collection was established by comparing the *XbaI*-PFGE profiles of the isolates obtained following the PulseNet protocol, and imported into BioNumerics (Applied Maths, St-Martens-Latem, Belgium) to perform a dendrogram with the UPGMA algorithm based on the Dice similarity coefficient and applying 1% of tolerance in the band position.

### 4.4. Genome Sequencing, Assembly and Analysis

DNA from 17 isolates was extracted with the QIAamp 96 DNA Qiacube HT kit (Qiagen, Hilden, Germany) and libraries were prepared using the Nextera XT kit (Illumina). Pooled libraries were denatured following the Illumina protocol and 600 μl (approx. 20 pM) were loaded onto a MisSeq V2 -500 cycle cartridge (Illumina) and sequenced on a MiSeq to produces fastq files. Raw reads were uploaded and automatically assembled in EnteroBase using SPAdes Genome Assembler v 3.5. with a contig threshold of minimum 200 nucleotides. Subsequently, the de novo assembled contigs were MLST (7 gene Achtman ST scheme, whole genome MLST, core genome MLST and ribosomal MLST) and serotyped in silico using EnteroBase typing tools [57]. The raw reads were also analyzed using the following CGE databases: SerotypeFinder⁠, MLSTtyper⁠, CHTyper⁠, PlasmidFinder, ResFinder, and VirulenceFinder [58,59,60,61,62]. For genomic relatedness comparison, we used different approaches based on the cgMLST of EnteroBase. Thus, a MSTree was inferred using the MSTree V2 algorithm and the asymmetric distance matrix based on the cgMLST scheme from EnteroBase. This cgMLST scheme consists of 2,513 genes present in over 98% of 3,457 genomes, which represented most of the diversity in EnteroBase https://enterobase.readthedocs.io/en/latest/pipelines/escherichia-statistics.html. We also investigated the HierCC designations for our collection and other related genomes of EnteroBase within each cluster group [57,63]. The SNP tree was also built in EnteroBase, where all assemblies were aligned against LREC-113 using Last [64], and SNPs from these alignments were filtered to remove regions with low base qualities or ambiguous alignment. Specifically, any sites with low base qualities (Q < 10) or sites which could not be aligned unambiguously (ambiguity of alignment ≥ 0.1, as reported by Last) were excluded. Additionally sites were removed if disperse repetitive regions were aligned with ≥ 95% identities and longer than ≥ 100 bps according to nucleotide BLAST; if they were part of tandem repeats that were identified by TRF [65]; if they were within CRISPR regions, which were identified by PILER-CR [66]. After removing repetitive regions, all core SNPs were then called in the core genomic regions that were conserved in ≥ 90% of the genomes.

## 5. Conclusions

In summary, our results demonstrate that a hybrid MDR aEPEC/ExPEC belonging to the clonal group O153:H10-A-ST10 (CH11-54) *eae*-beta1 is circulating in our region within different hosts, including wildlife. It seems implicated in human diarrhea via food (meat) transmission, and in the spreading of ESBL genes (mainly of CTX-M-32 type). The concomitant presence of IncF (F2:A-:B-), IncI1 (STunknown) and IncX1, together with non-conjugative Col156-like plasmids might be implicated in the successful persistence of this hybrid pathotype. We found genomic evidence of a related hybrid aEPEC/ExPEC in at least one other country.

## Data Availability

**The whole genome sequenced samples are part of BioProject PRJEB19190 and correspond to BioSample IDs:** SAMEA92137918 (LREC-110); SAMEA92139418 (LREC-111); SAMEA92142418 (LREC-112), SAMEA92143168 (LREC-113); SAMEA92149918 (LREC-114); SAMEA92149168 (LREC-115); SAMEA92148418 (LREC-116); SAMEA92146168 (LREC-117); SAMEA92154418 (LREC-118); SAMEA92147668 (LREC-119); SAMEA92144668 (LREC-120); SAMEA92146918 (LREC-121); SAMEA92151418 (LREC-122); SAMEA92150668 (LREC-123); SAMEA92152168 (LREC-124); SAMEA92152918 (LREC-125); SAMEA92140168 (LREC-127).

## Figures and Tables

**Figure 1 antibiotics-09-00192-f001:**
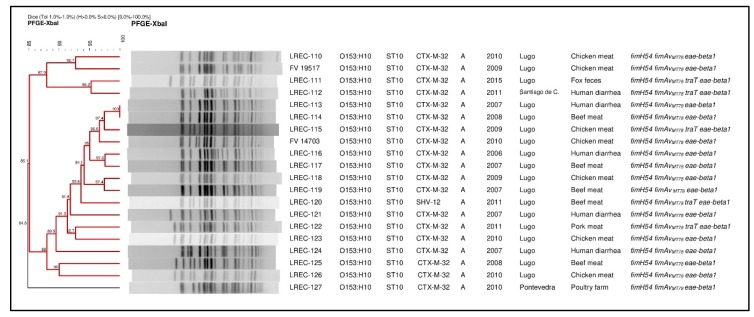
Pulsed field gel electrophoresis (PFGE) of *XbaI*-digested DNA from 20 ESBL-producing aEPEC isolates of the clonal group O153:H10-A-ST10 (one autodigested). On the right of the dendrogram: Isolate designation, O:H serotype, ST, ESBL type, phylogroup, year of isolation, geographic origin, source and virulence-gene profile.

**Figure 2 antibiotics-09-00192-f002:**
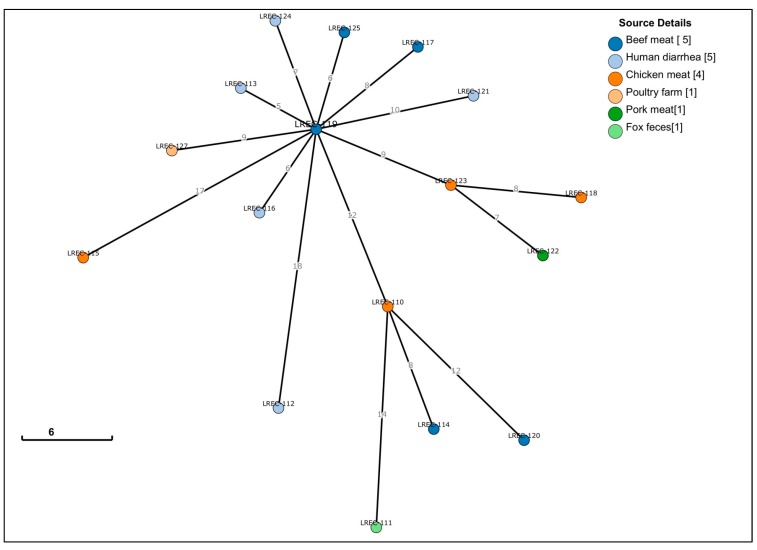
GrapeTree inferred using the MSTree V2 algorithm based on the cgMLST V1 + Hierarchical Clustering (HierCC) V1 scheme from EnteroBase.

**Figure 3 antibiotics-09-00192-f003:**
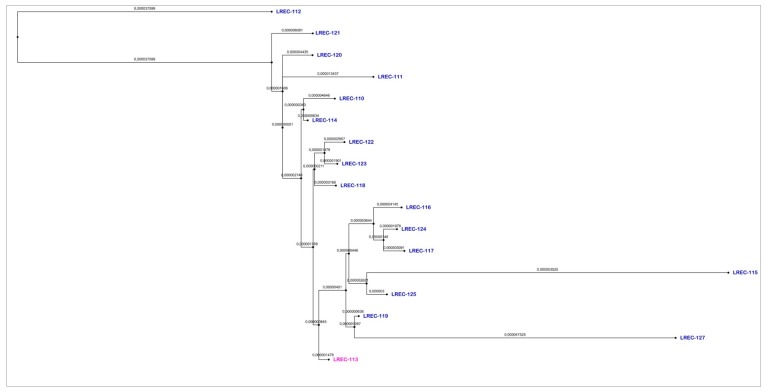
Dendrogram based on the SNPs of the core genomic regions present in 90% of the compared genomes and using LREC-113 as reference, built in EnteroBase and modified with FigTree v1.4.3.

**Table 1 antibiotics-09-00192-t001:** Phenotypic and genotypic characterization of 32 aEPEC O153:H10-A-ST10 (CH11-54) isolates.

Sample Origin	Code ^a^	Year	Geographic Origin	Virulence Gene Profile	Resistance Profile ^b^	*bla*_ESBL_ Type
Pork meat	*LREC-122	2011	Lugo	*fimH54, fimAv_MT78_, traT, eae-beta1*	AMP, CXM, CTX, CAZ, FEP, GEN	CTX-M-32
Chicken meat	*LREC-115	2009	Lugo	*fimH54, fimAv_MT78_, traT, eae-beta1*	AMP, CXM, CTX, CAZ, FEP, GEN, TOB	CTX-M-32
Chicken meat	FV 19517	2009	Lugo	*fimH54, fimAv_MT78_, eae-beta1*	AMP, CXM, CTX, CAZ, FEP, GEN	CTX-M-32
Chicken meat	*LREC-118	2009	Lugo	*fimH54, fimAv_MT78_, eae-beta1*	AMP, CXM, CTX, CAZ, FEP, GEN	CTX-M-32
Chicken meat	*LREC-110	2010	Lugo	*fimH54, fimAv_MT78_, eae-beta1*	AMP, CXM, CTX, CAZ, FEP, GEN	CTX-M-32
Chicken meat	FV 14703	2010	Lugo	*fimH54, fimAv_MT78_, eae-beta1*	AMP, CXM, CTX, CAZ, FEP, GEN, TOB, FOF	CTX-M-32
Chicken meat	LREC-126	2010	Lugo	*fimH54, fimAv_MT78_, eae-beta1*	AMP, CXM, CTX, CAZ, FEP, GEN, TOB	CTX-M-32
Chicken meat	*LREC-123	2010	Lugo	*fimH54, fimAv_MT78_, eae-beta1*	AMP, CXM, CTX, CAZ, FEP, GEN, TOB	CTX-M-32
Beef meat	*LREC-119	2007	Lugo	*fimH54, fimAv_MT78_, eae-beta1*	AMP, CXM, CTX, CAZ, FEP, GEN	CTX-M-32
Beef meat	* LREC-117	2007	Lugo	*fimH54, fimAv_MT78_, eae-beta1*	AMP, CXM, CTX, CAZ, FEP, GEN	CTX-M-32
Beef meat	4-3a	2007	Lugo	*fimH54, fimAv_MT78_, traT, eae-beta1*	AMP, CXM, CTX, CAZ	SHV-12
Beef meat	85-5a	2008	Lugo	*fimH54, fimAv_MT78_, traT, eae-beta1*	AMP, GEN	−
Beef meat	*LREC-125	2008	Lugo	*fimH54, fimAv_MT78_, eae-beta1*	AMP, CXM, CTX, FEP	CTX-M-32
Beef meat	*LREC-114	2008	Lugo	*fimH54, fimAv_MT78_, eae-beta1*	AMP, CXM, CTX, CAZ, FEP, GEN, TOB	CTX-M-32
Beef meat	65-6a	2009	Lugo	*fimH54, fimAv_MT78_, traT, eae-beta1*	−	−
Beef meat	*LREC-120	2011	Lugo	*fimH54, fimAv_MT78_, traT, eae-beta1*	AMP, CXM, CTX, CAZ, FEP	SHV-12
Wildlife (Fox)	*LREC-111	2015	Lugo	*fimH54, fimAv_MT78_, traT, eae-beta1*	AMP, CXM, CTX, CAZ, FEP, GEN, TOB	CTX-M-32
Poultry farm	*LREC-127	2010	Pontevedra	*fimH54, fimAv_MT78_, eae-beta1*	AMP, CXM, CTX, CAZ, FEP, GEN	CTX-M-32
Human	*LREC-116	2006	Lugo	*fimH54, fimAv_MT78_, eae-beta1*	AMP, CXM, CTX, CAZ, FEP, GEN, TOB	CTX-M-32
Human	*LREC-113	2007	Lugo	*fimH54, fimAv_MT78_, eae-beta1*	AMP, CXM, CTX, CAZ, FEP, GEN, TOB	CTX-M-32
Human	*LREC-121	2007	Lugo	*fimH54, fimAv_MT78_, eae-beta1*	AMP, CXM, CTX, CAZ, FEP, GEN, TOB	CTX-M-32
Human	*LREC-124	2007	Lugo	*fimH54, fimAv_MT78_, eae-beta1*	AMP, CXM, CTX, CAZ, FEP, GEN, TOB	CTX-M-32
Human	31952. 07	2007	Lugo	*fimH54, fimAv_MT78_, traT, eae-beta1*	−	−
Human	32651. 07	2007	Lugo	*fimH54, fimAv_MT78_, traT, eae-beta1*	NAL, CIP	−
Human	32884. 07	2007	Lugo	*fimH54, fimAv_MT78_, traT, eae-beta1*	AMP, CXM, CAZ, AMC, SXT	−
Human	34535. 07	2007	Lugo	*fimH54, fimAv_MT78_, traT, eae-beta1*	NAL, CIP	−
Human	39044. 07	2007	Lugo	*fimH54, fimAv_MT78_, traT, eae-beta1*	−	−
Human	21011. 08	2008	Lugo	*fimH54, fimAv_MT78_, traT, eae-beta1*	−	−
Human	38506. 08	2008	Lugo	*fimH54, fimAv_MT78_, traT, eae-beta1*	CIP	−
Human	40237. 08	2008	Lugo	*fimH54, fimAv_MT78_, traT, eae-beta1*	NAL, CIP	−
Human	*LREC-112	2011	Santiago de Compostela	*fimH54, fimAv_MT78_, traT, eae-beta1*	AMP, CXM, CTX, CAZ, FEP, NAL	CTX-M-32
Human	55515.12	2012	Lugo	*fimH54, fimAv_MT78_, traT, eae-beta1*	AMP, GEN	−

^a^ Strains marked with (*) were further analyzed by WGS; ^b^ ampicillin (AMP), amoxicillin/clavulanic acid (AMC), cefuroxime (CXM), ceftazidime (CAZ), cefotaxime (CTX), cefepime (FEP), cefoxitin (FOX), gentamicin (GEN), tobramycin (TOB), fosfomycin (FOF), sulfamethoxazole/trimethoprim (SXT), ciprofloxacin (CIP) and nalidixic acid (NAL).

**Table 2 antibiotics-09-00192-t002:** In silico characterization of 17 *Escherichia coli* genomes from the study collection using Center for Genomic Epidemiology (CGE) databases and ClermonTyping (in red, results obtained only by conventional typing).

Code	Serotype ^1^	Phylo Group ^2^	CHType ^3^	ST ^4^	Plasmid Content Inc Group (pMLST) ^5^	Acquired Resistances (black) and Point Mutations (blue) ^6^	Virulence Genes ^7,8^
LREC-110	O153:H10	A	11-54	10	IncF (F2:A-:B-) IncI1 (STunknown) IncX1 Col156	*bla_CTX-M-32_, bla_TEM-1A_; aac(3)-IIa, aadA1; catA1; mdf(A); tet(A)*	*astA, eae, espA, espB, espF, gad, iss, mchF, nleA, tccP, tir*
LREC-111	O153:H10	A	11-54	10	IncF (F2:A-:B-) IncI1 (STunknown) IncX1 Col156 Col (MG828)	*bla_CTX-M-32_, bla_TEM-1A_; aac(3)-IIa, aadA1; catA1; mdf(A); tet(A)*	*astA, eae, espA, espB, gad, iss, mchF, nleA, tccP, tir*
LREC-112	O153:H10	A	11-54	10	IncF (F2:A-:B-) IncX1 Col156 Col (MG828)	*bla_CTX-M-32__;_aadA1; catA1; mdf(A); tet(A); gyrA * S83L	*astA, eae, espA, espB, espF, gad, iss, mchF, nleA, tccP*
LREC-113	O153:H10	A	11-54	10	IncF (F2:A-:B-) IncI1 (STunknown) IncX1 Col156	*bla_CTX-M-32_, bla_TEM-1A_; aac(3)-IIa, aadA1; catA1; mdf(A); tet(A)*	*astA, eae, espA, espB, espF, gad, iss, mchF, tir*
LREC-114	O153:H10	A	11-54	10	IncF (F2:A-:B-) IncI1 (STunknown) IncX1 Col156 Col (MG828)	*bla_CTX-M-32,_ bla_TEM-1A_; aac(3)-IIa, aadA1; catA1; mdf(A); tet(A)*	*astA, eae, espA, espB, espF, gad, iss, mchF, nleA, tir*
LREC-115	O153:H10	A	11-54	10	IncF (F2:A-:B-) IncI1 (STunknown) IncX1 Col156	*bla_CTX-M-32_, bla_TEM-1A_; aac(3)-IIa, aadA1; catA1; mdf(A); tet(A)*	*astA, eae, espA, espB, espF, gad, iss, mchF, nleA, tccP, tir*
LREC-116	O153:H10	A	11-54	10	IncF (F2:A-:B-) IncI1 (STunknown) IncX1 Col156	*bla_CTX-M-32_, bla_TEM-1A_; aac(3)-IIa, aadA1; catA1; mdf(A); tet(A)*	*astA, eae, espA, espB, gad, iss, mchF, tccP, tir*
LREC-117	O153:H10	A	11-54	10	IncF (F2:A-:B-) IncI1 (STunknown) IncX1 Col156	*bla_CTX-M-32_; aadA1; mdf(A); tet(A)*	*astA, eae, espA, espB, gad, iss, mchF, tccP, tir*
LREC-118	O153:H10	A	11-54	10	IncF (F2:A-:B-) IncI1 (STunknown) IncX1 Col156 Col(MG828)	*bla_CTX-M-32,_ bla_TEM-1A_; aac(3)-IIa, aadA1; catA1; mdf(A); tet(A)*	*astA, eae, espA, espB, espF, gad, iss, mchF, nleA, tccP, tir*
LREC-119	O153:H10	A	11-54	10	Col156	*bla_CTX-M-32_* *, aadA1; catA1; mdf(A)*	*astA, eae, espA, espB, gad, iss, nleA, tccP, tir*
LREC-120	O153:H10	A	11-54	10	IncI1 (ST22-CC2) IncQ1 IncX1 Col156 Col (MG828)	*bla_SHV-12_; aadA1, aadA2; catA1, cmlA1; mdf(A); sul3; tet(A)*	*astA, eae, espA, espB, gad, iss, mchF, nleA, tccP, tir*
LREC-121	O153:H10	A	11-54	10	IncF (F2:A-:B-) IncI1 (STunknown) IncX1 Col156	*bla_CTX-M-32_, bla_TEM-1A_; aac(3)-IIa, aadA1; catA1; mdf(A); tet(A)*	*astA, eae, espA, espB, gad, iss, mchF, nleA, tccP, tir*
LREC-122	O153:H10	A	11-54	10	IncF (F2:A-:B-) IncI1 (STunknown) IncX1 Col156 Col (MG828)	*bla_CTX-M-32_; aac(3)-IIa, aadA1; catA1; mdf(A); tet(A)*	*astA, eae, espA, espB, gad, iss, mchF, nleA, tccP, tir*
LREC-123	O153:H10	A	11-54	10	IncF (F2:A-:B-) IncI1 (STunknown) IncX1 Col156 Col (MG828)	*bla_CTX-M-32_, bla_TEM-1A_; aac(3)-IIa, aadA1; catA1; mdf(A); tet(A)*	*astA, eae, espA, espB, gad, iss, mchF, nleA, tccP, tir*
LREC-124	O153:H10	A	11-54	10	IncF (F2:A-:B-) IncI1 (STunknown) IncX1 IncY Col156	*bla_CTX-M-32_, bla_TEM-1A_; aac(3)-IIa, aadA1; catA1; mdf(A); tet(A)*	*astA, eae, espA, espB, espF, gad, iss, mchF, tccP, tir*
LREC-125	O153:H10	A	11-54	10	IncF (F2:A-:B-) IncI1 (STunknown) IncX1 Col156	*bla_CTX-M-32_; aadA1; catA1; mdf(A); tet(A)*	*astA, eae, espA, espB, espF, gad, iss, mchF, nleA, tccP, tir*
LREC-127	O153:H10	A	11-54	10	IncF (F2:A-:B-) IncI1 (STunknown) IncX1 Col156 Col (MG828)	*bla_CTX-M-32_, bla_TEM-1A_; aac(3)-IIa, aadA1; catA1; mdf(A); tet(A)*	*astA, eae, espA, espB, espF, gad, iss, mchF, nleA, tccP, tir*

^1^ Serotypes, ^3^ clonotypes, ^4^ sequence types, ^5^ replicon/plasmid STs, ^6^ acquired antimicrobial resistance genes and/or chromosomal mutations, ^7^ virulence genes were determined using SerotypeFinder 2.0, CHtyper 1.0, MLST 2.0, PlasmidFinder 2.0, pMLST 2.0, ResFinder 3.1 and VirulenceFinder 2.0 online tools at the CGE, respectively. While ^2^ phylogroups were predicted using the ClermonTyping tool at the Iame-research Center web. ^1^ Serotypes: underlined and in red those (LREC-121, LREC-120) that were not predicted (ONT) by SerotypeFinder but assigned as O153 by conventional typing. ^6^ Resistome: Acquired resistance genes: beta-lactam: *bla*_TEM-1A_, *bla*_CTX-M-32,_
*bla*_SHV-12_; aminoglycosides: *aac(3)-IIa, aadA1, aadA2;* phenicols: *catA1, cmlA1;* macrolides: *mdf(A)*; sulphonamides: *sul3*; tetracycline: *tet(A).* Point mutations (marked in blue): quinolones and fluoroquinolones: *gyrA* S83L: TCG-TTG. Underlined and in red those *bla*_CTX-M-32_ genes (LREC-112, LREC-119) that were not predicted by ResFinder but determined in conventional typing ^8^ Virulence genes: *astA*: EAST-1, *eae*: intimin, *espA*: type III secretions system, *espB*: secreted protein B, *espF*: type III secretions system, *gad*: glutamate descarboxylase, *iss*: increased serum survival, *mchF*: ABC transporter protein MchF, *nleA*: non-LEE-encoded effector A, *tccP*: Tir cytoskeleton coupling protein, *tir*: translocated intimin receptor protein. bp: base pairs; CHType: clonotype (*fumC*-*fimH*); ST: sequence type according to Achtman scheme; pMLST: plasmid sequence type.

**Table 3 antibiotics-09-00192-t003:** Asymmetric distance matrix based on the cgMLST scheme from EnteroBase in which D (a, b) equals all sites that are present in (b) and different from (a).

Genome Code/ cgMLST		LREC- 110	LREC- 111	LREC- 127	LREC- 112	LREC- 113	LREC- 120	LREC- 117	LREC- 121	LREC- 119	LREC- 116	LREC- 115	LREC- 114	LREC- 123	LREC- 122	LREC- 124	LREC- 125	LREC- 118
		37600	37601	37602	37605	37606	37607	37609	37610	37611	37612	37613	37614	37615	37616	37617	37618	38299
LREC-110	37600	0	14	17	19	13	12	18	19	12	15	27	8	15	15	18	16	14
LREC-111	37601	14	0	21	23	16	20	22	23	16	19	30	11	19	18	22	20	17
LREC-127	37602	17	21	0	24	9	22	14	16	9	11	24	15	13	13	15	13	13
LREC-112	37605	19	23	24	0	19	25	24	25	18	21	33	17	22	22	24	23	21
LREC-113	37606	13	16	9	19	0	18	9	11	5	6	18	10	9	8	11	9	8
LREC-120	37607	12	20	22	25	18	0	23	23	18	20	33	14	20	20	24	22	19
LREC-117	37609	18	22	14	24	9	23	0	17	8	12	22	15	14	13	14	12	13
LREC-121	37610	19	23	16	25	11	23	17	0	10	13	25	15	16	16	16	14	14
LREC-119	37611	12	16	9	18	5	18	8	10	0	6	17	10	9	9	7	6	8
LREC-116	37612	15	19	11	21	6	20	12	13	6	0	22	12	11	11	13	11	10
LREC-115	37613	27	30	24	33	18	33	22	25	17	22	0	22	24	23	23	15	20
LREC-114	37614	8	11	15	17	10	14	15	15	10	12	22	0	13	12	15	14	11
LREC-123	37615	15	19	13	22	9	20	14	16	9	11	24	13	0	7	15	13	8
LREC-122	37616	15	18	13	22	8	20	13	16	9	11	23	12	7	0	15	13	9
LREC-124	37617	18	22	15	24	11	24	14	16	7	13	23	15	15	15	0	12	14
LREC-125	37618	16	20	13	23	9	22	12	14	6	11	15	14	13	13	12	0	12
LREC-118	38299	14	17	13	21	8	19	13	14	8	10	20	11	8	9	14	12	0

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
