# Peer review of "Genomic Characterization of Escherichia coli Isolates Belonging to a New Hybrid aEPEC/ExPEC Pathotype O153:H10-A-ST10 eae-beta1 Occurred in Meat, Poultry, Wildlife and Human Diarrheagenic Samples"

_antibiotics, 2020, doi:10.3390/antibiotics9040192_

Round 1

Reviewer 1 Report

Review April 2020

I read with interest this manuscript. The manuscript reports the molecular characterization of unusual serotype of E. coli and it shows its dissemination along 10 years and among different samples collected from food-producing animals, humans and one fox, a serotype also reported in other countries. First, I think the manuscript would improve by a revision of English language.

The data reported in the manuscript is scientifically interesting and it adds some information to the state of art on E. coli enteropathogenic, namely in its epidemiology and molecular features of virulence and antimicrobial resistance. Nonetheless, I found that introduction should be more detailed and more focused in this particular serotype (see my comment below). Actually, the title does not express the aim of the study; this study does not focus only in ESBL E. coli producers. Moreover, most of the isolates carrying the blaCTX-M-32, which is relevant (only one SHV-12 and many human samples with no CTX-M-32). In the discussion, some of my questions about the methods are answered. Overall, I do not think the conclusions fit the objectives described previously for the study or that the data obtained in this study support the view elaborated by the authors. However, the work and the data presented is valuable and interesting for many readers.

Below, authors can find my questions/comments/ suggestions:

Introduction

Line 46. No specific VF for each sub-type… how UPEC is characterized?

line 61 “through”.

Line 63. Bacteriophages contribute for the genetic evolution of bacteria but are not MGEs. They are virus.

Lines 71-76. During 10 years authors recovered 21 ESBL positive of this serotype. Which is the relevance of this serotype? Introduction does not say anything about it. What is known about this serotype? The objectives of the study are enumerated but which is the relevance of this study? Introduction should have more information to understand the final propose of the study.

Another question: the title focused in ESBL positive strains but this not the focus of study according the objectives. The authors also included non-ESBL producers.

Line 93. … producers (or positive). Here, authors could highlight some data that come from the table, such as, most of human samples were ESBL negative. TEM enzymes were not amplified? (It is common in E. coli TEM1/2, though not ESBL)

Discussion

Line 264-…. I do not think that your results support the spread of CTX-M-32. Or maybe, it is not explained very well what authors want to say. From my point of view, it is the strain that is circulating for years among animals and humans. Interestingly, many of the human E,coli isolates do not produce CTX-M-32. Is it the same plasmid that is circulating with blaCTX-M-32? If the plasmids were identical and the isolates genetically different, so the ESBL could be spreading among different strains. It seems more interesting to highlighting the virulence of this strain that may reach humans via food-chain, and that this knowledge is important to implement measures to contain the spread of pathogenic E. coli.

Moreover, the last sentence should be reviewed. This study does not prove the presence of the hybrid isolate in other countries. It proves that this serotype is circulating in different countries, namely in northwest of Spain, and this might be due to different reasons (including animal trade).

Methods

Line 276: What environmental samples and from wildlife?

Line 278. CIT is an AmpC beta-lactamase. Despite a broad substrate range is not classified as an ESBL (like carbapenemases). Why only two groups of CTX-M beta-lactamases?

Line 282. Are those VF specific of EXPEC?

During 10 years authors only found 32 eae-positive E. coli? If not, why did they focus in this serotype?

Line 300. This sentence is not clear “Sequencing of the specific regions was performed for conventional typing of blaBLEE genes (Table S9)”. Please, describe the methodology with more detail.

Line 300. Why did authors use CLSI guidelines when EUCAST rules are followed, namely for human samples, in Europe Union countries?

Line 310. What was the base of selection of 17 isolates to sequence?

Author Response

We thank all comments from reviewer 1 that helped to improve the manuscript. Thus, the English language has been reviewed. More information is included in the introduction section. The title and abstract have been changed. Besides, the objectives and conclusions have been rewritten. Please find below our point-by-point answers.

Introduction

  1. Line 46. No specific VF for each sub-type… how UPEC is characterized?

This point has been clarified now (Lines 66-74), including more information about ExPEC:

E. coli that cause extraintestinal infections are referred to as extraintestinal pathogenic E. coli (ExPEC), which includes a heterogeneous group of pathotypes defined by isolation from infections outside the intestinal tract: uropathogenic E. coli (UPEC), avian pathogenic E. coli (APEC), and neonatal meningitis E. coli (NMEC). A variety of specific virulence genes have been described in ExPEC, namely, adhesins, protectins, siderophores, toxins, hemolysins, polysaccharide antigens, invasins, colicins, etc. But even though certain virulence traits1 and phylogenetic groups have been proposed to characterize ExPEC, no set of genes can be used to unequivocally distinguish them from commensal E. coli. In fact, ExPEC strains can commensally colonize the human intestine, which in turn, can serve as reservoir”

1As additional explanatory note for reviewer: there are useful markers that predict the potential virulence of the isolates to cause an extraintestinal infection (doi: 10.1128/AAC.47.7.2161-2168.2003) and, specifically, a uropathogenic infection. Thus, Spurbec et al (2012) found that Escherichia coli isolates that carry vat, fyuA, chuA, and yfcV efficiently colonize the urinary tract (doi: 10.1128/IAI.00752-12). Since ExPEC are a heterogeneous group of pathogens that encompasses avian, neonatal meningitis, and uropathogenic E. coli strains, and there were no core set of virulence factors that can be used to definitively differentiate these pathotypes, the authors analyzed four virulence factor-encoding genes, yfcV, vat, fyuA, and chuA, highly associated with uropathogenic E. coli strains. Their results indicated that a predictor gene (PG) score of 3 or 4 of those is indicative of UPEC, and isolates with a PG score of 4 may be highly virulent.

  1. line 61 “through”.

Thank you, it has been corrected (Line 78).

  1. Line 63. Bacteriophages contribute for the genetic evolution of bacteria but are not MGEs. They are virus.

Bacteriophages and phage-related particles have recently been highlighted as MGEs that transfer antibiotic resistance (https://doi.org/10.1016/j.plasmid.2015.01.001).

What we meant in this sentence is that the horizontal gene transfer represents a key mechanism by which naturally occurring antibiotic resistance genes captured by a mobile scaffold such as an integron, transposon, phage, plasmid, or chromosomal island move from an environmental source into clinically relevant bacterial species (in this case, E. coli) 10.3389/fmicb.2013.00086.

Based on above, we´ve decided to maintain the sentence.

  1. Lines 71-76. During 10 years authors recovered 21 ESBL positive of this serotype. Which is the relevance of this serotype? Introduction does not say anything about it. What is known about this serotype? The objectives of the study are enumerated but which is the relevance of this study? Introduction should have more information to understand the final propose of the study.

Thank you for your comments. Now, we have tried to explain the objectives and relevance of the study better (Lines 89-97): “Different in-house surveillance studies in northwest Spain (2005-2015) revealed the presence of eae-positive isolates of E. coli O153:H10, many of them ESBL-producers, in meat for human consumption, wildlife, and avian farm environment. We also found them involved in human diarrhea. This study aimed the genetic and genomic characterization of representatives to determine the degree of relatedness between human and animal/meat isolates, as well as to gain knowledge in the mechanism that might be playing a role in its persistence. We also wanted to know if it was a geographically restricted E. coli lineage, or reported somewhere else.”

As additional explanatory note for reviewer: The detection of this hybrid pathotype was based on the following findings: First, we noticed the recovery of eae-positive isolates with the same intimin type. Second, many of them were ESBL-producers (coming from surveillance studies on ESBL-producing E. coli in animal sources). Third, they belonged to the O153 serogroup. Fourth, we found them within the human collection in the retrospective analysis. Fifth, the isolates also carried ExPEC virulence genes. This information has been also included in the revised version (Lines 102-106).

We have included information on the serogroup O153 in the Discussion section (Lines 240-243). While O153 is a serogroup reported within ExPEC and DEC isolates, and linked to different lineages, there are few references of the serotype O153:H10. Thus, it was reported by Schremmer et al. associated to an aEPEC isolation from the small intestine of a cockatiel (psittaciform bird) with enteritis

Therefore, we had not been really following this hybrid until we were conscious of its presence, and it is difficult to summarize all this information in a manuscript (we´ve tried our best now). We confirm that we are still detecting it within new studies (in progress). In summary, the serotype is an additional trait of this clonal group. For us, the importance here is to investigate what makes certain clonal groups successfully congregate different virulence/resistance traits. It is also crucial to report the presence / emergence of them, considering previous events such as that of the Shiga-toxin-producing E. coli (STEC) belonging to serotype O104:H4, with virulence features (VF) common to the enteroaggregative E. coli (EAggEC ), and CTX-M-15 producer, involved in the large German outbreak.

  1. Another question: the title focused in ESBL positive strains but this not the focus of study according the objectives. The authors also included non-ESBL producers.

The new title is:“Genomic characterization of Escherichia coli isolates belonging to a new hybrid aEPEC/ExPEC pathotype O153:H10-A-ST10 eae-beta1 occurred in meat, poultry, wildlife and human diarrheagenic samples

  1. Line 93. … producers (or positive). Ok. It has been done as suggested (Line 124).

Here, authors could highlight some data that come from the table, such as, most of human samples were ESBL negative. TEM enzymes were not amplified? (It is common in E. coli TEM1/2, though not ESBL)

We cannot estate that most of human samples were ESBL negative. As explained above, most isolates of animal origin came from ESBL surveillance studies, while human isolates were retrospectively recovered for this study from the routine diagnosis collection (not focuses on ESBLs). So, we cannot really compare the prevalence and evolution of this clonal group, concerning ESBL-gene presence, within human and animal isolates.

As for the TEM enzymes, they were amplified but only ESBLs appear in Table 1. You are right; TEM-1 is very common in E. coli but not ESBL. Table 2 shows it was predicted (blaTEM-1) in 11 of the 17 genomes.

Discussion

Line 264-…. I do not think that your results support the spread of CTX-M-32. Or maybe, it is not explained very well what authors want to say. From my point of view, it is the strain that is circulating for years among animals and humans. Interestingly, many of the human E,coli isolates do not produce CTX-M-32. Is it the same plasmid that is circulating with blaCTX-M-32? If the plasmids were identical and the isolates genetically different, so the ESBL could be spreading among different strains. It seems more interesting to highlighting the virulence of this strain that may reach humans via food-chain, and that this knowledge is important to implement measures to contain the spread of pathogenic E. coli.

Moreover, the last sentence should be reviewed. This study does not prove the presence of the hybrid isolate in other countries. It proves that this serotype is circulating in different countries, namely in northwest of Spain, and this might be due to different reasons (including animal trade).

Please, see comments above.

From our point of view, the importance of our study is I) the report of a new hybrid pathotype present in different sources; II) its association with MDR and specifically CTX-M-32; III) the genomic relatedness between isolates of animal (including meat for human consumption) and human origin.

While the serotype is interesting it is not a definitive trait at all, since it could be determined within other different and not related pathotypes. At the same time, there is a high degree of recombination in the genetic region coding O antigen in E. coli (differently from H antigen). In fact, we found other similar and genomic related isolates in Enterobase with different O antigen, as shown in Table S6.

We believe that this new version of the manuscript explains better all this information. The conclusions of the study have been revisited addressing reviewer´s comments (Lines 312-318). “In summary, our results demonstrate that a hybrid MDR aEPEC/ExPEC belonging to the clonal group O153:H10-A-ST10 (CH11-54) eae-beta 1 is circulating in our region within different hosts, including wildlife. It seems implicated in human diarrhea via food (meat) transmission, and in the spreding of ESBL genes (mainly of CTX-M-32 type). We found genomic evidence of a related hybrid aEPEC/ExPEC in, at least, another country”.

Methods

Line 276: What environmental samples and from wildlife?

Ok, this information has been added (Line 326). They were avian feces taken from the floor and fox feces, respectively.

Line 278. CIT is an AmpC beta-lactamase. Despite a broad substrate range is not classified as an ESBL (like carbapenemases). Yes, you are right. It was also checked by PCR but none of the isolates was positive.

Why only two groups of CTX-M betalactamases? Because they comprises practically 100% of the ESBL-types found in our isolates.

Line 282. Are those VF specific of EXPEC? Yes, they are specific virulence genes that have been described in ExPEC, even though no set of genes can be used to unequivocally distinguish them (please, see explanation in the Introduction section).

During 10 years authors only found 32 eae-positive E. coli? If not, why did they focus in this serotype?

Please, check answer 4.

Line 300. This sentence is not clear “Sequencing of the specific regions was performed for conventional typing of blaBLEE genes (Table S9)”. Please, describe the methodology with more detail.

Ok, the information is included now. It means that the isolates were investigated by PCR for screening of specific bla genes using the TEM, SHV, CTX-M-1 and CTX-M-9 group-specific primers, and further sequencing (sanger sequencing) as described elsewhere (Mora et al., 2013). Table S9 shows the specific primers, conditions and references.

Line 300. Why did authors use CLSI guidelines when EUCAST rules are followed, namely for human samples, in Europe Union countries? I do not have a categorical answer, apart from it is widely used in scientific publications (comparison reason).

Line 310. What was the base of selection of 17 isolates to sequence?

Ok, this is now explained in Line 143: “Based on the high similarity shown by PFGE and to further investigate the virulence profile, resistome, plasmid content and relatedness, 17 representative aEPEC/ExPEC isolates were WG sequenced.

Reviewer 2 Report

The presented manuscript based on genomic characterization of ESBL-producing E. coli is an interesting and well-written manuscript. This study would be useful for the readers of the “Antibiotics” journal after addressing the following minor comments.

  1. In Section 2.1 please briefly define PFGE.
  2. In section 2.2 please define the WgMLST and cgMLST briefly.
  3. Table 2 some rows showing serotype and genes in red font. what's red font mean?
  4. In discussion line 4 presenting 5.5.% 20% two values for beef meat. which one is correct?
  5. In methodology section 4.2 please provide a reference for "Disc diffusion assay".

Author Response

Thank you very much. We appreciate reviewer´s comments.

  1. In Section 2.1 please briefly define PFGE.

Ok. Now, it has been defined (Line 129) as pulsed field gel electrophoresis (PFGE).

  1. In section 2.2 please define the WgMLST and cgMLST briefly.

Ok. Now, they have been defined (Lines 153-156) as follows: “Whole genome multilocus sequence typing (WgMLST) and core genome multilocus sequence typing (cgMLST) are powerful schemes with extreme and high resolution, respectively. cgMLST is defined as MLST based on the core genome, whereas wgMLST is based on a non-redundant set of genes across a species, similar to a ‘pan-genome’. In Enterobase, the number of core loci included in the cgMLST scheme for E. coli is 2,512, and 25,002 in the wgMLST”

  1. Table 2 some rows showing serotype and genes in red font. what's red font mean?

It means that the result shown was obtained only by conventional typing, but not predicted in silico. This was already explained in Line 191 for serotypes and in Line 193 for bla genes. Now, it is also indicated in the headline of table 2 (Line 187).

  1. In discussion line 4 presenting 5.5.% 20% two values for beef meat. which one is correct?

Thank you for the correction, 20% is the right value (Line 215).

  1. In methodology section 4.2 please provide a reference for "Disc diffusion assay".

Ok. A reference has been included and the sentence modified as follows (Lines 345-346 and 350-351): “Antimicrobial susceptibility testing was conducted by disk (Becton Dickinson, Sparks, MD, USA) diffusion assay….. The assays were performed, and all results interpreted, according to the CLSI guidelines.

Yours sincerely,

Azucena Mora

------------------------------------------------------------------------------
Azucena Mora Gutiérrez, DVM, PhD

Associate Professor
Laboratorio de Referencia de E. coli (LREC)

Department of Microbiology and Parasitology

Faculty of Veterinary Sciences,

Universidad de Santiago de Compostela (USC)

27002- Lugo (Spain)

Phone: +34 982 822110
e-mail: azucena.mora@usc.es
-----------------------------------------------------------------------------------
